



# Sedimentary microplankton distributions are shaped by oceanographically connected areas

Peter D. Nooteboom[1,2], Peter K. Bijl[3], Christian Kehl[1], Erik van Sebille[1,2], Martin Ziegler[3], Anna S. von der Heydt[1,2], and Henk A. Dijkstra[1,2]

[1]Institute for Marine and Atmospheric research Utrecht (IMAU), Department of Physics, Utrecht University
[2]Centre for Complex Systems Studies, Utrecht University
[3]Marine Palynology and Paleoceanography, Laboratory of Palaeobotany and Palynology,Department of Earth Sciences, Utrecht University

**Correspondence:** P.D.Nooteboom (p.d.nooteboom@uu.nl)

**Abstract.** Having descended through the water column, microplankton in ocean sediments are representative for the ocean surface environment, where they originated from. Sedimentary microplankton is therefore used as an archive of past and present surface oceanographic conditions. However, these particles are advected by turbulent ocean currents during their sinking journey. So far, it is unknown to what extent this particle advection shapes the microplankton composition in sediments. Here we use global simulations of sinking particles in a strongly eddying global ocean model, and define ocean bottom provinces based on the particle surface origin locations. We find that these provinces can be detected in global datasets of sedimentary microplankton assemblages, demonstrating the effect provincialism has on the composition of sedimentary remains of surface plankton. These provinces explain the microplankton composition, together with e.g. ocean surface environment. Connected provinces have implications on the optimal spatial extent of microplankton sediment sample datasets that are used for palaeoceanographic reconstructions, and on the optimal spatial averaging of sediment samples over global datasets.

## 1 Introduction

Microplankton communities are sensitive to surface oceanographic conditions in which they live. Their remains are preserved in the sedimentary archive of the ocean basins and are therefore used to reconstruct present and past surface ocean conditions. However, the sedimentary microplankton community is not driven by abiotic climate variables (e.g. temperature or nutrient availability) alone. These climate variables only explain part of the sedimentary species variability, for both dinoflagellate cysts (Zonneveld et al., 2010; Esper and Zonneveld, 2007) and planktic foraminifera (Morey et al., 2005). As a result, there is a large unexplained residual error in relationships between plankton composition and environmental conditions. This impacts accuracy of reconstruction of past environmental conditions using microfossil allemblages.. Hence, it is crucial to investigate which other processes determine the species distribution in the sedimentary archive, especially when such distributions are used to reconstruct these sea surface variables in the geologic past.

Global surface-ocean currents, and the way in which these currents connect the ocean, are shown to shape the plankton community structure near the ocean surface (Jonnson and Watson, 2016; Wilkins et al., 2013; Hellweger, 2014). Connectivity





of the two-dimensional (2D) surface ocean flow is well-studied in models (Froyland et al., 2007, 2014; Onink et al., 2019; McAdam and van Sebille, 2018). Floating particles accumulate towards the so-called garbage patches on decadal time scales

(Lebreton et al., 2012), which often match well with relatively high concentrations of surface drifters (van Sebille et al., 2012) and microplastics (van Sebille et al., 2015b). In addition to the ocean surface connectivity, the three-dimensional (3D) ocean connectivity is expected to have an influence on the distribution of sedimentary particles.

Recent studies show that advection of sinking particles in 3D ocean flow has implications for sedimentary microplankton distributions (van Sebille et al., 2015a; Nooteboom et al., 2019). An inititally uniform distribution of particles at the ocean

surface becomes more heterogeneous (i.e. mixed) when these particles are sinking (Monroy et al., 2019; Drótos et al., 2019). At the same time, the influence of ocean currents on sedimentary particle distributions is spatially varying (Nooteboom et al., 2020). Hence, one might expect that the sedimentary archive is shaped by 3D particle advection by ocean currents during the sinking process.

The behavior of sinking particles in a 3D flow is quite different compared to that in a 2D flow. For instance, a 2D flow is

divergent in an upwelling region (which drives the particle convergence in garbage patches), while a 3D flow is non-divergent. However, attracting structures (Bettencourt et al., 2012) and transport barriers (Bettencourt et al., 2015; Chang et al., 2018) of 3D particle paths can also emerge inside the ocean. In this way, particles can cluster in specific areas when they are collected at a 2D surface after their sinking journey (Monroy et al., 2017; Eaton and Fessler, 1994), as is also measured at the ocean subsurface (Mitchell et al., 2008; Logan and Wilkinson, 1990).

In this paper, we investigate how oceanographically disconnected areas shape the sedimentary microplankton composition. We cluster sedimentary sites based on similar ocean surface origin locations of particles that ended up at these sediment sites after their sinking journey. We compare the clusters with well-known features of the ocean flow and detect these clusters in measurements of sedimentary microplankton.

## 2   Method

### Sedimentary data

We use two global datasets of sedimentary microplankton, one with dinoflagellate cysts (dinocysts; Marret et al., 2019) and the second with planktic foraminifera (Siccha and Kucera, 2017). We use the surface sediment samples from sites South of $65°N$ (2849 and 4017 sites for the dinocysts and foraminifera respectively), because the OFES ocean model (which is used for particle advection; see below) ends at $75°N$, which makes the clustering results at high Northern latitudes unreliable. For some

statistical analyses, we only consider sites in the Southern Hemisphere (725 and 1858 sites for respectively the dinocysts and foraminifera), in order to limit the total diversity of microplankton species in the datasets. We consider the fraction (i.e. the relative abundance) of microplankton species for every surface sediment sample.





**Clustering methods and particle tracking**

The particle tracking results from Nooteboom et al. (2019) provide us with distributions of surface origin locations for a global
$1° \times 1°$ grid of sediment sites, for several sinking speeds. We use the results that are obtained in the eddying OFES model
(Sasaki et al., 2008; Masumoto et al., 2004) with a sinking speed of 6 m day$^{-1}$. Results with a sinking speed of 11, 25 and
250 m day$^{-1}$ can be found in the Supporting Information. Values of 6 and 11 m day$^{-1}$ are representative of single sinking
dinocysts, 25 m day$^{-1}$ for small aggregates, and 250 m day$^{-1}$ for large aggregates and planktic foraminifera (Anderson et al.,
1985; Nooteboom et al., 2019, 2020; van Sebille et al., 2015a).

The sinking speeds and backtracking analysis from Nooteboom et al. (2019) are specifically designed to be compatible with
the life cycle of dinocysts (Nooteboom et al., 2019): particles are released at the bottom of the ocean every 5 days, and tracked
back in time until they reach 10 m depth, providing a particle distribution at the ocean surface (Fig. 1a). Single foraminifera
typically sink at higher velocities than dinocysts ($\gtrsim 100$ m day $^{-1}$), and most of their lateral transport occurs during their life
span, when they are passively advected while they control their buoyancy and remain at their preferential depth (van Sebille
et al., 2015a). However, we assume in this paper that the strength, direction and 'mixing' of planktic foraminifera by ocean
currents has a similar spatially varying character compared to sinking dinocysts. We test whether the clustering results match
both dinocyst and foraminifera sample datasets.

Our goal is to obtain provinces of sediment sites from the back-tracked surface origin locations which are oceanographically
(i) *disconnected* (i.e. provinces between which particles are not likely to travel) and (ii) *isolated* (i.e provinces with sediment
sites which share similar origin locations compared to the sediment sites outside of the province). We quantify these areas
by disconnected and isolated clusters of sedimentary sites. Assuming that the flow from 2000 to 2005, as simulated by the
OFES model, is representative of the real ocean flow in the past decades (during which the microplankton actually sedimented;
Jonkers et al., 2019), we ideally find the disconnectedness and isolation of clusters in the surface sediment sample datasets.

We use two types of clustering techniques. First, hierarchical clustering provides boundaries where sinking particles are less
likely to cross (hence it finds oceanographically disconnected areas). This technique starts with the full ocean as only cluster
and splits a cluster into two clusters at every iteration (see Appendix A1 for more details). The clusters from this technique
can be compared to areas that are known to be oceanographically (dis)connected from each other, and these clusters can be
used to test if more similar microplankton species are found within each connected area compared to between connected areas.
Advantages of the hierarchical clustering method are that the cluster structure is preserved as more iterations are applied, and it
does not require many parameters. The only parameter that the hierarchical clustering uses is the stop-criterion (i.e. the iteration
number where the algorithm stops with creating new clusters).

Second, we use the *Ordering Points To Identify the Clustering Structure* (OPTICS) algorithm to find oceanographically
isolated clusters. OPTICS provides a density based value (the reachability) of sedimentary sites which quantifies how strongly
a site is connected to other sites. Oceanographically isolated clusters can be obtained from the 'dense regions' (i.e. areas with
low reachability values), by setting a threshold on the slope that surrounds the dense values in the reachability plot ($\xi$; see Fig.
1b for an example). The sediment sites outside of these clusters are less isolated, and refered to as 'noisy.' These clusters allow



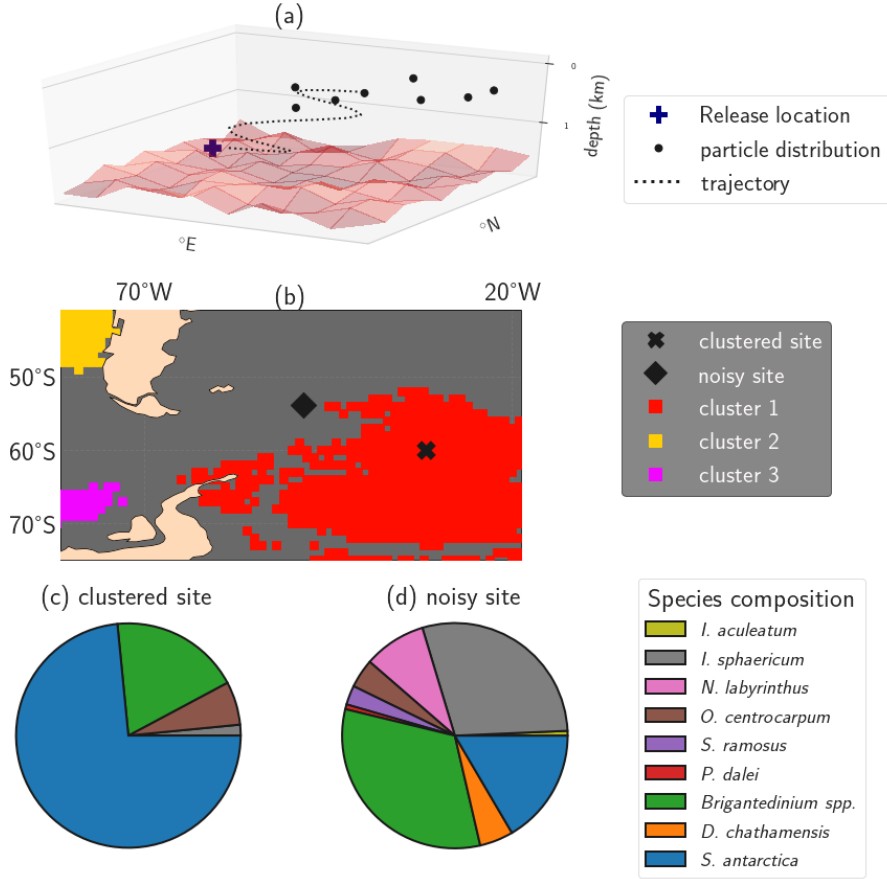

**Figure 1.** Illustration of the impact of isolated clusters on sedimentary microplankton composition. (a) Illustration of the particle back-track analysis from Nooteboom et al. (2019), resulting in a particle distribution of origin locations for one sediment site/release location on which the clustering methods are applied (figure adapted from Nooteboom et al. (2020))). (b) A (noisy) sediment sample site outside of the isolated clusters (station J299) and a site within oceanographically isolated OPTICS cluster 1 (station J285) in the South Atlantic. (c), (d) Pie charts of the dinocyst species composition in the sites from (b). The clustered site contains a species composition which is less biodiverse compared to the noisy site. The Shannon biodiversity indices of respectively the clustered and noisy site are 0.7788 and 1.6842. This illustration uses the same OPTICS clusters as are shown later in Fig. 4.

us to test if sedimentary species compositions are more homogeneous inside isolated areas compared to outside of these areas (see Fig.1).

The advantage of OPTICS is that parameter values have a clear interpretation. First, the parameter $s_{min}$ is the minimum number

of particle release locations in clusters, which represents a minimum spatial scale of clusters (in m$^2$). The second parameter $\xi$ determines the degree of isolation of the clustering: OPTICS generally finds less and smaller clusters if $\xi$ is larger. Another



advantage of OPTICS is that not all areas are clustered, such that it allows to distinguish between 'noisy' (not clustered) and oceanographically isolated (clustered) areas (see Appendix A2 for more details about OPTICS).

We also test the seasonal dependence of hierarchical clusters (see Supporting Fig. 3 and 4), by only considering particles
that started sinking in a specific season (in the backtracking analysis of Fig. 1a). While some of the cluster boundaries changed between summer and winter, the change in the overall clustering structure was limited if only a specific season of origin locations was considered (similar to van Sebille et al. (2015a), who only found a small seasonal effect in temperature offsets due to lateral transport of foraminifera).

**Statistical analyses**

We apply several statistical tools to test hypotheses about the sediment sample sites and the clusters in which they are located. A partial Mantel test (Legendre and Legendre, 2012) is used to test whether the reachability from the OPTICS algorithm correlates with the sediment sample taxonomy, independent of the spatial distance between sediment sample site locations. A partial Mantel test requires at least three types of distance matrices, which contain distances between the sediment sample sites. We calculate the Mantel correlation between taxonomic distance and a distance which is determined from the reachability of
the OPTICS clustering (see Appendix B), while we control for the spatial distance between sites.

We use Canonical Correspondence Analysis (CCA; Braak and Verdonkschot (1995)) to infer the relation between species in clustered sediment sites and environment parameters at the ocean surface. In this context, CCA ideally shows unique species responses to changes in environment input parameters. We use sea surface temperature (SST) and surface nitrate ($NO_3$) as environmental parameters, as prior literature reported them to explain a major part of the species variability in the Southern
Ocean (Prebble et al., 2013; Esper and Zonneveld, 2007). This study infers SST and $NO_3$ for sediment sample locations from $1° × 1°$ fields of the *World Ocean Atlas* (Locarnini et al., 2013; Garcia et al., 2013). Further parameters, such as phosphorus, silicate, salt concentration, were tested (as in Hohmann et al. (2019)), though spurious CCA response led to their exclusion from further analysis in this paper.

We compare the CCA's explained variation of sedimentary samples only drawn from (i) only isolated clusters and (ii)
with samples drawn from all available locations. Comparing the explained species variation of both cases allows us to draw conclusions about the source of variation between both CCA results in order to quantify the significance of the clustering approach. We apply a one-sided randomization test to investigate whether the increase of explained variance is significant. This implies that we randomly take subsamples of the full dataset, which are equally sized to the amount of clustered sediment samples. The p-value of the permutation test is the fraction of random subsamples that resulted in a higher explained variance
compared to the CCA analysis with the clustered samples.

The foraminifera dataset also contains deep dwelling species, which live near the thermocline (typically a few 100 meters depth). Although it is often assumed that these deep dwelling species relate to sea surface variables in statistical analyses, this assumption might not be valid (Telford and Kucera, 2013). We applied the CCA analysis while only using the species which are known to be near-surface dwelling in the subtropical Atlantic (the red group in figure 7 of Rebotim et al. (2017); supporting





Fig. 11 in this paper). This leads to similar conclusions, although less significant values are obtained because the dataset size
is lower.

The clusters that are obtained from the OPTICS algorithm represent areas of relative oceanographic isolation. We test
whether the species distributions in sediments outside clusters are more mixed compared than samples inside clusters during
their sinking journey. We use Shannon entropy (Shannon, 1948) to quantify taxonomic mixing, which is defined at site $j$

as $N_s^j = -\sum_i p_{ij} \ln(p_{ij})$. Here $p_{ij}$ denotes the relative abundance of a species $i$ at site $j$. Shannon entropy is often used as a
biodiversity index (Morris et al., 2014), being a combined signal of species richness (amount of species in the sediment sample)
and evenness (how evenly these species are distributed). We choose the Shannon entropy here as biodiversity index, because it
can be compared to the mixing of sinking particles, and Shannon entropy is often used to quantify the loss of information by
mixing (e.g. in thermodynamics). We compare the average Shannon entropy of sediment sample sites within ($\overline{N}_s^c$) and outside

($\overline{N}_s^{nc}$) clusters.

## 3   Results

### Oceanographically disconnected clusters

We interpret splits of oceanographically disconnected clusters from the hierachical clustering method as boundaries with a low
connectivity across them (Fig. 2). The probability that particles cross these boundaries is larger if the iteration number is higher.

We observe that these cluster edges compare well to large-scale ocean connectivity. The first iteration splits the Mediteranean
Sea from the global ocean, because no sinking particles travel through the Gibraltrar strait in the simulation. Next, the Pacific
is separated from the Arctic, since few particles are transported through Bering strait (Coachman and Aagaard, 1988). At
the subsequent iterations, the large-scale ocean basins disconnect: The Pacific, Atlantic and Indian ocean are split from the
Southern ocean at approximately 25°S. We find that areas near Western boundary currents may only split into clusters at

relatively high iterations, because the sediments in these areas have a relatively large connectivity, with particles originating
from a large area (see also Nooteboom et al. (2019)).

In the North-Atlantic region, we observe that Hudsons Bay becomes a cluster and the subtropical Atlantic is split from the
Nordic seas at the Greenland-Scotland ridge (Fratantoni, 2001; McClean et al., 2002; Bower et al., 2019). The Irminger Basin is
still connected with the Labrador Sea, where sinking of water occurs (Pickart et al., 2003; Katsman et al., 2018), which makes

transport of sinking particles outside of this area less likely. Only a few particles cross the connection between the Labrador
Sea and Baffin Bay (McClean et al., 2002; Fischer et al., 2018).

We do not find a cluster at subpolar latitudes which isolates Antarctica along latitudinal bands (only the Weddell and the
Ross Sea are a cluster). One might expect such a cluster, because near-surface currents are known to isolate Antarctica (Fraser
et al., 2018; Dufour et al., 2015; Doos et al., 2008). However, deep passive particles advected by three-dimensional flow are

shown to move upwards along isopycnals towards Antarctica (Drake et al., 2018; Tamsitt et al., 2018). As a result, the sinking
particles can be transported towards Antarctica at depth. The location of southward particle transport is mainly determined by



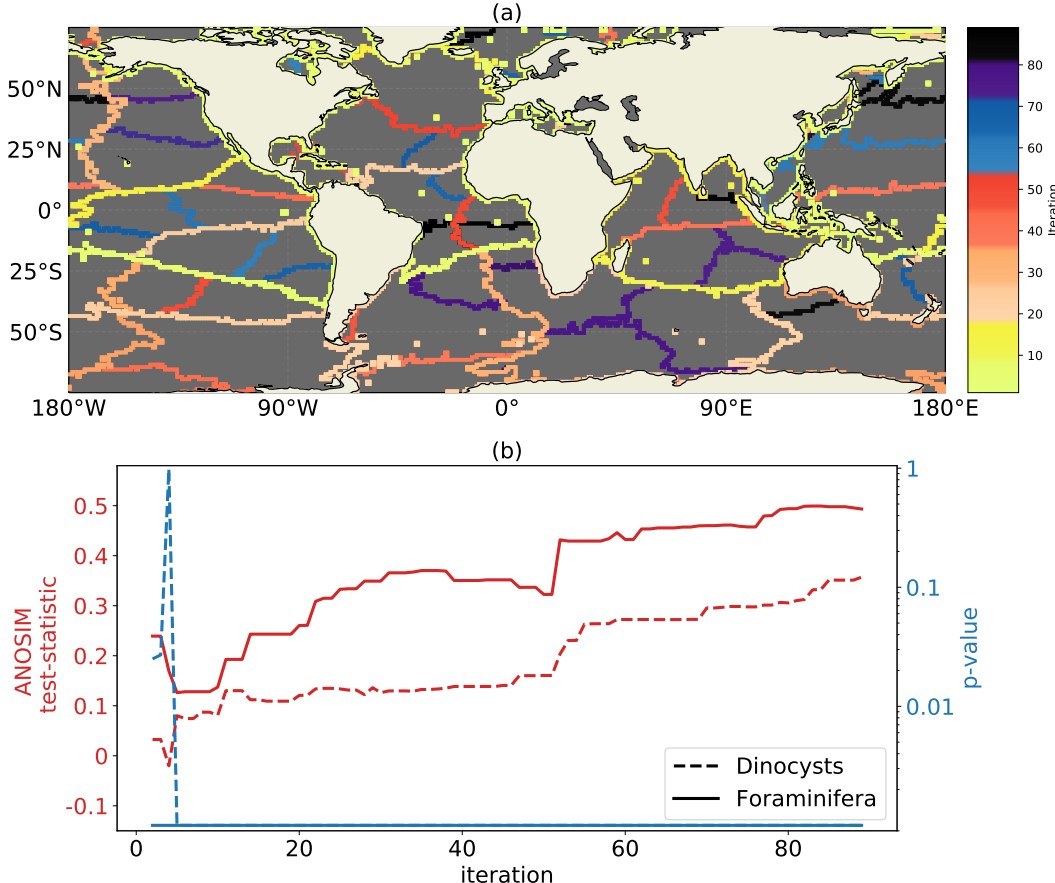

**Figure 2.** Edges between oceanographically disconnected clusters of sedimentary locations from the hierarchical clustering method. (a) The cluster edges after 90 iterations, where the color indicates at which iteration number a cluster edge is created. (b) The ANOSIM test-statistic (red) and p-values (blue; 999 permutations; logarithmic scale) for the clusters at every iteration number, which tests whether the sedimentary microplankton composition (both dinocysts and foraminifera; sites below 65°N) is more similar within than between clusters.

topographic steering of the flow, resulting in five hotspots of southward particle transport (Tamsitt et al., 2017) which roughly coincide with the Southern Ocean clusters in Fig. 2.

Some of the clusters in Fig. 2 are similar to connected regions based on the surface flow (see figure 8 from Froyland et al. (2014)). The North and South Atlantic are split similarly from West-Africa to Venezuela. The North Pacific and South Pacific are split in a similar way from Australia to the south of Chile. A cluster around the Pacific cold tongue (East Tropical Pacific) develops (Moum et al., 2013; Froyland et al., 2014). Moreover, the Benguela upwelling area (Nelson and Hutchings, 1983) (near South West Africa) is more connected with the Southern Ocean than with the Atlantic. Near-surface currents have an important influence on the total lateral transport of sinking particles in these areas, since they are similar to the surface connectivity areas from (Froyland et al., 2014).





We test whether sites within hierarchical clusters have a lower (Euclidean) taxonomic distance compared to sites of different clusters with *Analysis of similarities* (ANOSIM; Clarke (1993)). The clustering corresponds to a high statistical signigicance for (i) positive ANOSIM test-statistic together with (ii) low p-values (p-value<0.001). According to the ANOSIM tests, the clustering of sediment samples is significant across all iterations (Fig. 2c). Hence, sediment samples within clusters are more similar than those between clusters. The test-statistic increases at higher iteration numbers, for both the dinoflagellate cyst (dinocyst) and foraminifera dataset. Although these ANOSIM results look promising, it is important to note that the ANOSIM test-statistics are partly positive because the sediment sites within clusters are closer to each other (i.e. there is a distance effect independent of the clustering).

The hierarchical clustering is overall insensitive to the used sinking speed of particles (see the Supporting Information Fig. 1 and 2 with sinking speeds 11 and 25 m day$^{-1}$ respectively; see Appendix A1 for an explanation on why we did not test a sinking speed of 250 m day$^{-1}$). Only some minor differences occur in the North Pacific, and some cluster separations occur at slightly different iteration numbers. The fact that similar boundaries of little cross-transport emerge at a different sinking speed proves that the clustering does not greatly depend on the sinking speed of particles.

**Oceanographically isolated clusters**

The OPTICS clustering algorithm provides a density based value (the reachability) of sedimentary sites, which quantifies how strongly a site is connected to other sites. Oceanographically isolated clusters can be obtained from the 'dense regions' (i.e. areas with low reachability values), by setting a threshold on the slope that surrounds the dense values in the reachability plot ($\xi$). The sediment sites outside of these clusters are less isolated, and refered to as 'noisy.'

According to the OPTICS algorithm, the Western boundary currents are unlikely to be part of any isolated cluster, since the points near the Western boundary currents have a relatively high reachability (Fig. 3a). This is expected, because the origin locations of sediment samples near Western boundary currents comprise a large area. Dense areas are those at higher latitudes, close to Antarctica and in the Nordic Seas, and the midlatitude gyres. Sediment sample sites within these areas have more similar surface origin locations compared to sediment sites outside of dense areas. The high reachability values in the Mediterranean Sea and Red Sea are rather artificial. OPTICS searches for dense regions (low reachability) by searching for sedimentary sites with relatively many other sites having similar surface origin locations. Since the Mediterranean Sea and Red Sea are enclosed by land, these sedimentary sites have only few neighbouring sites, resulting in a relatively high reachability.

The reachability distance ($D^r$; see Appendix B) between sediment sample sites correlates positively with sediment sample taxonomy. Furthermore, it is independent of the spatial distance between sites, according to the partial Mantel tests (Fig.3c). Large values of $s_{min}$(>600; i.e. the OPTICS parameter which determines the minimum surface area in m$^2$ of OPTICS clusters) tend to have the largest correlation (also at other sinking speeds; see Appendix B). This is probably because the reachability is smoother at higher $s_{min}$, which makes the reachability distance less noisy. At small spatial scales ($s_{min} \leq 200$), the correlation between dinocyst taxonomy and $D^r$ could be indirect, because then $D^r$ correlates more strongly with the environment (in terms of sea surface temperature) compared to taxonomy.





**Figure 3.** Reachability plot of the sedimentary particle release locations from the OPTICS algorithm. Sediment locations in dense areas (i.e. with low reachability values in (a)) share a similar particle distribution of back-tracked surface origin locations, while areas with high reachability values have back-tracked particle distribution which are more spread out and share origin locations with a lot of other sedimentary release locations. A sinking speed of 6 m day$^{-1}$ is used and parameter $s_{min} = 300$ (i.e. OPTICS clusters will consist of a minimum of 300 sediment sites). (a) Scatter plot of the site reachability in space: sites in dense areas with a low reachability are oceanographically isolated. (b) A scatter plot of the ordering of the sediment locations $i$ against their reachability $r(p_i)$. (c) Partial Mantel correlation of the reachability distance $D^r$ with the taxonomy (red) and SST (black), both with spatial distance held constant, for different $s_{min}$ values. A total of 999 permutations were used for every partial Mantel test; every test with respect to the taxonomy (red) is significant with p-value$< 0.003$.

We compute clusters by setting a threshold on the slope ($\xi$) that surrounds the reachability valleys in Fig. 3a. For $\xi = 0.002$
and $s_{min} = 300$ (Fig. 4), we obtain 13 clusters, of which three regions are isolated by the Antarctic Circumpolar Current (ACC), three in the Indian ocean, one in the Pacific warm pool, the South Atlantic gyre, near the Humboldt upwelling zone,



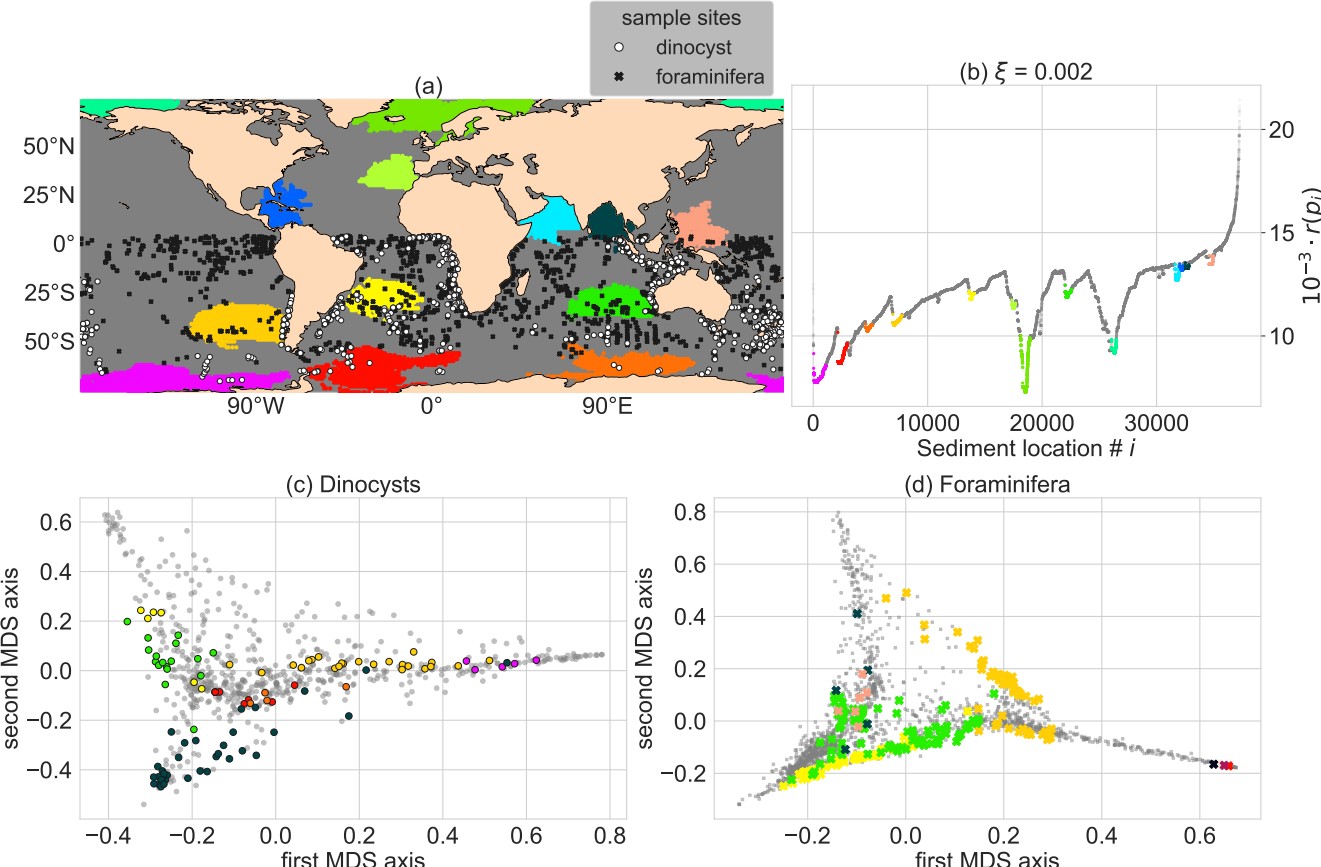

**Figure 4.** Oceanographically isolated OPTICS clusters of sedimentary particle release locations with clustering parameters $s_{min} = 300$ (i.e. the minimum size of clusters) and $\xi = 0.002$ (i.e. the level of isolation). The clustering is applied globally and the clusters are compared to Southern Hemisphere sediment sample sites. The colored regions are clusters, the gray regions are "noisy", and therefore not part of a cluster. These colors were used for all subpanels. (a) Global map of the position of the clusters (colored regions), and dinocyst- (white) and foraminifera (black) sample locations. (b) Ordering of sedimentary locations $i$ against their reachability $r(p_i)$. To visualize the sediment sample site taxonomy for (c) the dinocysts and (d) the planktic foraminifera in two dimensions, we use *classical multidimensional scaling* (MDS; Fouss et al. (2016)). MDS creates a two-dimensional approximation of the species composition in the sediment samples in this figure (instead of 91 and 50 dimensions/species for the dinocysts and foraminifera respectively).

near the Caribean Sea, the Eastern North Atlantic and two clusters near the Arctic. The clusters represent locations that are oceanographically isolated, with sediment sample sites that have backtracked origin locations which are similar to the other sites in the cluster. The environmental variability within these clusters can be reasonably large (e.g. the sea surface temperatures

at backtracked origin locations in the cluster West of Australia range between 10-25°C; see planktondrift.science.uu.nl).

The comparison between the clusters and taxonomic distance of Southern Hemisphere sample sites in these clusters (Fig. 4c and 4d) becomes interesting for clusters which are spatially close (Fig. 4b). For instance, the red and yellow cluster in





the South-Atlantic Ocean are spatially close, but sediment samples in those clusters are separated by their observed dinocysts taxonomy (Fig. 4c). This implies that we find a signal of the oceanographic separation of these areas in the sedimentary data.

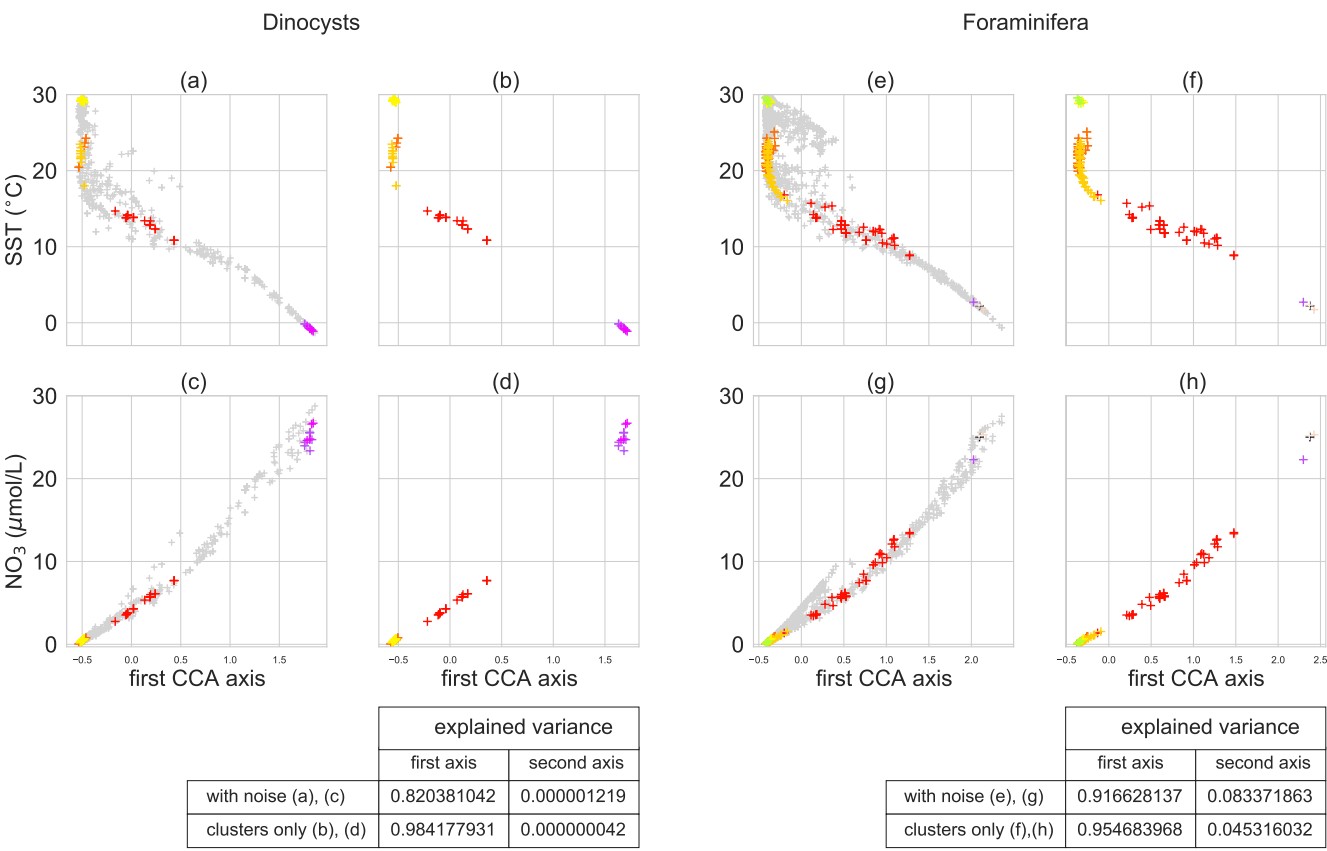

| | explained variance | |
| --- | --- | --- |
| | first axis | second axis |
| with noise (a), (c) | 0.820381042 | 0.000001219 |
| clusters only (b), (d) | 0.984177931 | 0.000000042 |

| | explained variance | |
| --- | --- | --- |
| | first axis | second axis |
| with noise (e), (g) | 0.916628137 | 0.083371863 |
| clusters only (f),(h) | 0.954683968 | 0.045316032 |

**Figure 5.** The relation between microplankton species variability and environmental variables according to a CCA analysis, while including and excluding unclustered sediment samples (using the isolated clusters from Fig.4). Sea surface temperature (top) and nitrate concentration (bottom) at sediment sample sites with dinocysts (left; (a), (b), (c), (d)) and foraminifera (right; (e), (f), (g), (h)) against the first canonical axis from the CCA analysis, including ((a),(c),(e),(g)) and excluding ((b),(d),(f),(h)) sediment sample sites outdside of the oceanographically isolated clusters. The sediment sample sites that belong to a cluster are colored, 'noisy' samples (i.e. not part of any cluster) are gray. The tables at the bottom show the proportion of total variance that is explained by the canonical axes if the noisy samples are included or excluded. 13.5% (for dinocysts) and 10.8% (for foraminifera) of the sediment sample sites is in clusters, the remainder is in 'noisy' regions. The increase of explained variance is supported by a permutation test with 999 permutations (p-values are <0.0001 and 0.024 for dinocysts and foraminifera respectively).

To test if sedimentary sites within clusters are better correlated with environmental conditions at the surface, we applied CCA either including or excluding the sedimentary sites outside the isolated clusters (Fig.5). We find that the amount of explained variation by the canonical axes increases significantly if noisy sediment samples are excluded for the foraminifera





(∼0.92 to ∼0.95) and especially for the dinocysts (∼0.82 to ∼0.98), for the same OPTICS clusters as in Fig.4. Hence, we find

that the linear relationship between environmental variables and microplankton composition of the CCA explains a larger part

of the sedimentary species compisition if noisy sites are excluded. In that sense, the signal is 'cleaner' for sediment sample

sites within compared to outside clusters, which has implications for palaeoceanographic reconstructions of SST with these

sedimentary data.

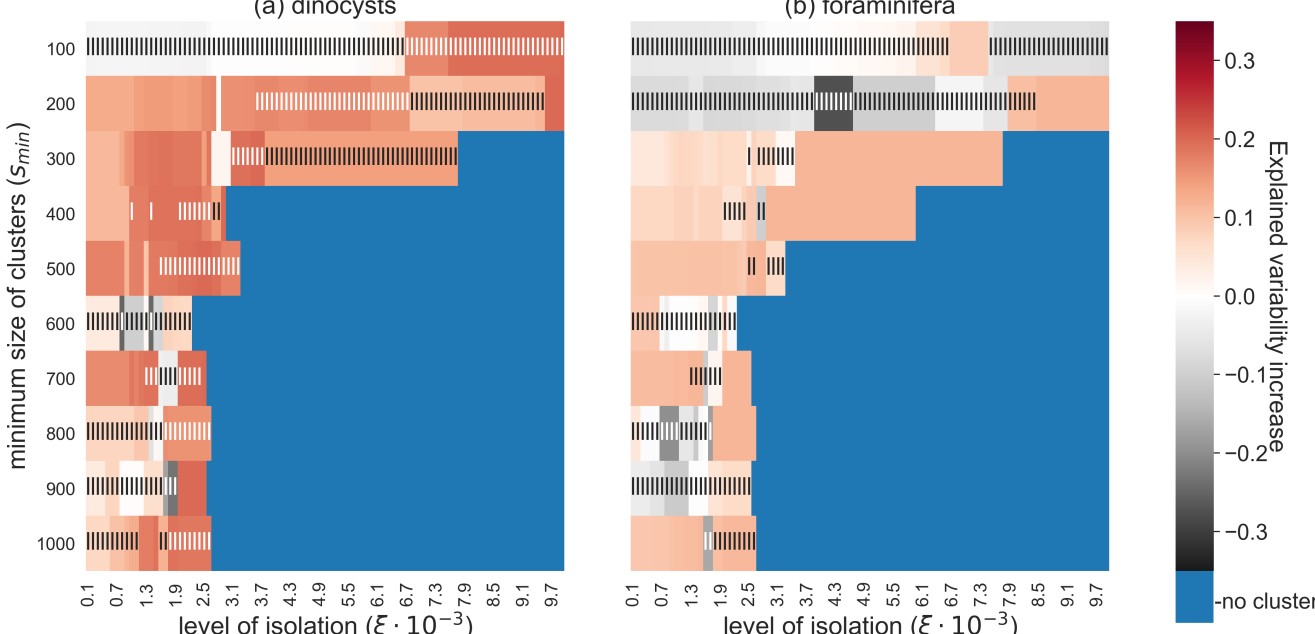

**Figure 6.** Increase of explained environmental variability by microplankton sediment sample sites in the CCA analyses if sediment samples

outside of the oceanographically isolated OPTICS clusters are excluded, for different parameter values $s_{min}$ (i.e. the minimum size of

clusters) and $\xi$ (i.e. the level of isolation). (a) the dinocyst and (b) the foraminifera dataset. High and significant values indicate that sediment

samples within clusters have a clearer relationship with the surface environment. Blue are configurations of $s_{min}$ and $\xi$ for which no sediment

sample sites are part of a cluster. Only Southern Hemisphere sedimentary microplankton data were used here. Vertical stripes indicate an

insignificant randomization test with 999 permutations at a 5% significance level.

The robustness of the relationship between sedimentary sites and environmental variables is investigated by testing the

sensitivity of the CCA results to these parameters $\xi$ and $s_{min}$ (Fig. 6). For the dinocyst dataset we find an increase of explained

variation by the canonical axes for most tested values of $s_{min}$ and $\xi$. By increasing the reachability slope that surrounds the

oceanographically isolated clusters ($\xi$), a higher constraint is put on the isolation of these clusters and less sediment sample

sites are part of a cluster. If this slope $\xi$ is chosen too high, no clusters exist or they are too small to contain any sediment

sample sites at all. Moreover, a higher value of $\xi$ means that less sedimentary sites are used in the CCA (i.e. the dataset size

is reduced), which may lead to an insignificant result according to the randomization test. A relatively low value of $\xi$ on the



other hand, may lead to insignificant results due to the inclusion of noisy sites in clusters. Hence, there seems to be an optimal

value $\xi$, for which this increase of variance is maximized. These results are more often insignificant for the dinocysts compared

to the foraminifera, because the dinocyst dataset is smaller. If $s_{min}$ is higher (i.e. the OPTICS algorithm finds larger clusters;

in $km^2$), the negative and insignificant values in Fig. 6 are partly caused by including noisy sites in clusters. These results

highlight the importance of choosing an appropriate combination of $\xi$ and $s_{min}$ for the CCA to show a significant increased

explained species variability if only clustered sites are used.

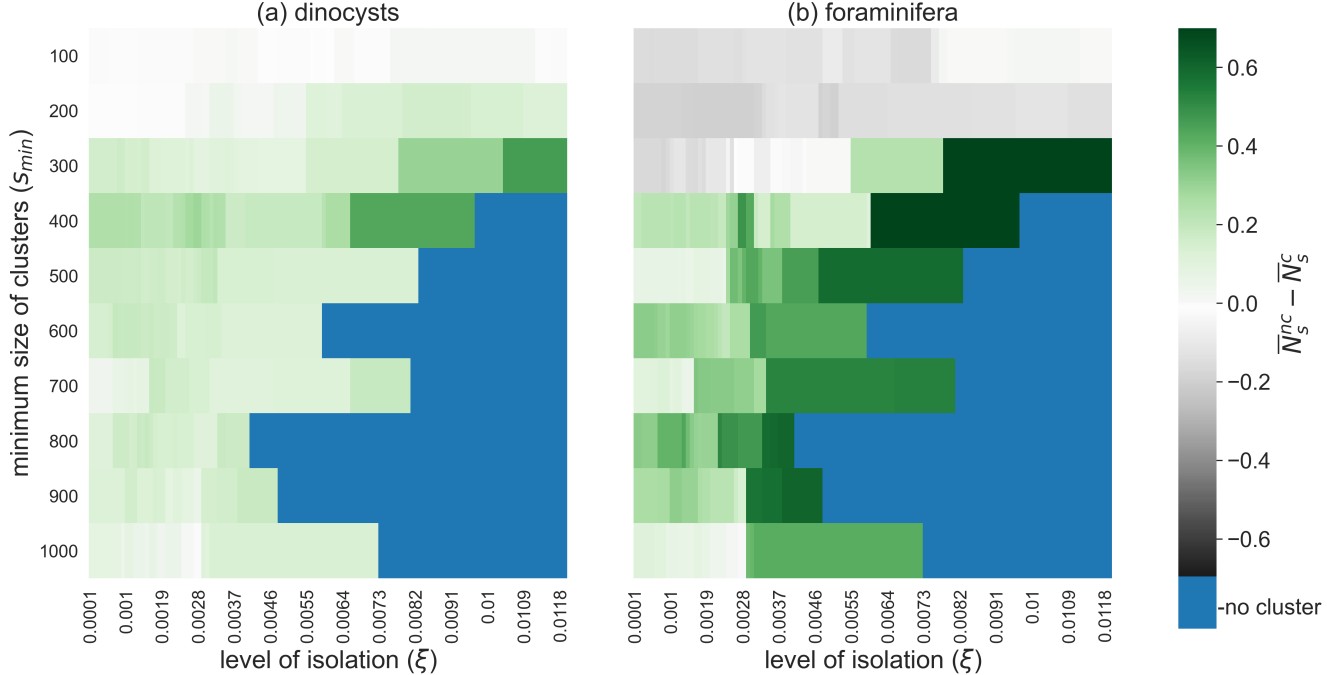

**Figure 7.** Sedimentary microplankton biodiversity outside minus inside isolated OPTICS clusters. The average Shannon entropy of sediment samples inside OPTICS clusters $\overline{N}_s^c$ compared to outside clusters $\overline{N}_s^{nc}$ for different values of $s_{min}$ (i.e. the minimum size of clusters) and $\xi$ (i.e. the level of isolation). High values indicate that the number of species in samples within clusters are lower and species are distributed less evenly in samples compared to samples outside clusters. Blue are configurations of $s_{min}$ and $\xi$ for which no sediment sample sites are part of a cluster.

The clustered samples are less taxonomically mixed (i.e. are less biodiverse) for most values of $\xi$ and $s_{min}$, as we can see

from the comparsion between the Shannon entropy ($N_s$; see method section) within and outside of clusters (Fig. 7). For the

high values of $s_{min}$, we find that the clusters are less taxonomically mixed if $\xi$ is increased. This result supports that measured

microplankton biodiversity in sediments is relatively large in areas with strong mixing of sinking particles by ocean currents.

However, the Shannon entropy is also influenced by the species distributions at the ocean surface, for which much less species

composition data is available compared to their sedimentary remains. At smaller values of $s_{min}$ ($s_{min} < 200$ and $s_{min} < 300$

for the dinocysts and foraminifera respectively; Fig. 7) also high (surface) productivity areas are clustered (e.g. the South-West



Atlantic or the Humboldt area; see Fig. 4). Hence, a relatively high sedimentary biodiversity in these clusters can be explained by the high biodiversity at the ocean surface, before these particles start sinking.

We also tested the OPTICS results for sinking velocities higher than 6 m day$^{-1}$ (11, 25 and 250 m day$^{-1}$; see Figures S5-S10 and S11-S14 in the Supporting Information). Similar clusters can be found with the other sinking velocities. A higher sinking speed decreases the particle travel time, hence the lateral transport and the mixing of sinking particles is overall lower (Nooteboom et al., 2019). However, the spatial dependence of the lateral transport is similar: both at low and high sinking velocities, the lateral particle transport is relatively large near Western boundary currents, and low in the middle of midlatitude

gyres (Nooteboom et al., 2020). As a result, the clusters are located in similar areas for different sinking speeds. It is only the spatial scale of these clusters that might be different. The spatial scale (i.e. the size) of the clusters can again be controlled by the parameters $s_{min}$ and $\xi$. Note that high productivity areas are less likely clustered at higher sinking speeds, which has implications for the Shannon entropy within and outside clusters (Fig. 7): Higher biodiversity is measured outside compared to within clusters for a larger area of $s_{min}$ and $\xi$ values as the sinking speed increases.

## 250    4    Discussion

We clustered sediment sites based on the ocean surface origin locations of sinking particles that end up at these sites in an ocean model. These clusters reveal which sedimentary areas are oceanographically (i) (dis)connected or (ii) isolated. The connectivity which is given by the clusters is an aggregate of the ocean connectivity at all depths that the sinking particles traverse before ending up in the sediments. This type of connectivity, and the way it shapes the sedimentary microplankton composition, is

additive to the environment and surface ocean connectivity which influences the plankton community structure at the ocean surface (Jonnson and Watson, 2016; Wilkins et al., 2013). Nevertheless, the near-surface flow likely has a large imprint on the clusters since these contain the strongest ocean currents.

It was shown before, that the ocean surface ecological affinity of certain sedimentary microplankton species can improve if the lateral advection of sinking particles is taken into account (Nooteboom et al., 2019). This paper explores reasons why

sedimentary plankton assemblages include species that occur outside their surface water habitat range. Microplankton species mix by turbulent ocean currents during their sinking journey, which can result in a relevant lateral displacement along transport. The extent by which this occurs differs strongly in the world oceans, and is larger in areas that are referred to as noisy in this paper.

We conclude that ocean sediments are to a spatially varying degree provincial, and province boundaries are governed by

near-surface and deep currents in the ocean. These provinces have an impact on sedimentary microplankton assemblages. Their quantification helps to determine ocean sediment regions that are oceanographically (a) (dis)connected and (b) isolated from the area outside of these regions. Quantification of connected and isolated provinces have at least 4 implications for future studies.

First, the clustering methods that are presented in this paper can help to improve the application of transfer functions on

microplankton assemblages. Transfer functions train a model on surface sediment samples and ocean surface environmental



variables (in the present-day), in order to make quantitative climate reconstructions of past climates from microplankton in deeper sediments. Hence, these transfer function models use spatial variability of an environmental variable to predict its temporal variability in a single location. One challenge of transfer functions is to choose a proper spatial extent to train the prediction model (Hohmann et al. (2019); often in the present-day situation). A small spatial extent does not capture enough
of species and environment variability. If the spatial extent is too large, different processes determine the sedimentary species distribution which reduces the transfer function skill.

The hierarchical clustering method (which finds oceanographically disconnected clusters) can help to determine bounds on the spatial extent that is used for the training of transfer functions, since it shows areas which are oceanographically separated from each other. These clusters are created in a present-day configuration in this study, and may change in past climates. The
OPTICS clustering can be used to find oceanographically isolated clusters to determine the spatial extent of a regional transfer function model. In this case, it is advisable to check if the OPTICS cluster is large enough (i.e. the deep sediments do not contain species outside of the cluster).

Second, the connectivity between provinces could have an effect on biogeochemical properties of microplankton species that are applied as a proxy of the ocean surface environment. Hence, these provinces can be used to correct for ocean connectivity
if the proxies are used to assimilate e.g. global sea surface temperature fields (as in Tierney et al. (2020)). Moreover, spatially varying Bayesian regression is used to some of these biogeochemical proxies, because the proxy response differs across oceanic basins (Tierney and Tingley, 2015, 2018). The disconnected provinces from this paper could provide a spatial structure that such a regression model uses for core-top calibration.

Third, the results in this paper have implications for other types of sinking particles in the ocean. For instance, a large fraction
of marine plastic sinks to the ocean floor (Canals et al., 2020; Kooi et al., 2017). Sedimentary plastic distributions might be subject to similar mechanisms of mixing during their sinking journey. Clustering of sedimentary sites might indicate where the largest inhomogenities of sedimentary plastics appear (de la Fuente et al., 2021), or boundaries where sinking plastic is less likely to cross.

Fourth, our study provides micropalaeontologists with a tool to qualitatively assess the importance of lateral transport to
sedimentary particle assemblages, which can be used in studies that compare measured biological diversity and environmental conditions in surface waters with their sedimentary remains (e.g. Jonkers et al. (2019); Meilland et al. (2020)), particularly in those regions for which we here demonstrate 'noisy' behaviour. These type of studies could determine the relative contribution to the higher biodiversity outside compared to within oceanographically isolated clusters from ocean surface parameters, as well as dissolution (Frenger et al., 2018; Taylor et al., 2018) and mixing of particles during their sinking journey.
Drivers of biodiversity at the ocean surface, such as species interactions (Lima-Mendez et al., 2015), ecological limits and evolutionary dynamics (Fenton et al., 2016), are complex. It is possible that oceanographically isolated provinces do not directly drive a low biodiversity, but indirectly cause a low biodiversity through a lower variation of abiotic factors. Moreover, these provinces are likely located in areas with relatively little eddy activity, while mesoscale eddies can explain relatively high biodiversity values (Frenger et al., 2018).




The backtracking analysis on which we applied the clustering was designed for dinocysts, and not for foraminifera. Clustering results compared well with the foraminifera dataset in most cases, because the areas with strong particle mixing and lateral transport (i.e. their spatial dependence) are likely similar for foraminifera. Nevertheless, future work could apply these clustering methods on a backtracking analysis which is designed for foraminifera (similar to van Sebille et al. (2015a); Lange and Sebille (2017)). This means that particles are released at the ocean bottom, tracked back in time until they reach the

foraminifera dwelling depth, and finally tracked back during their life span at this dwelling depth.

*Code availability.* The code used for this work and the results are distributed under the MIT license and can be found at the website https://github.com/pdnooteboom/ClusterSinkingParticles.

*Data availability.* The data with back-tracked origin locations can be accessed on planktondrift.science.uu.nl. The datasets with measured dinocysts and planktic foraminifera data are from Marret et al. (2019) and Siccha and Kucera (2017) respectively.

**Appendix A: Clustering methods**

**A1   Hierarchical clustering**

The particle tracking results from Nooteboom et al. (2019) can be described by a bipartite graph (or transportation matrix; Fig A1a). This bipatite graph consists of bottom and surface nodes (representative of surface and bottom boxes in the transportation matrix; we use $1° \times 1°$ boxes in this paper). Bottom and surface nodes are linked if the probability that a particle which is found

in a bottom box originates from the surface box is greater than zero. We will use the projection of this bipartite graph on the bottom nodes (Fig 1c). This provides us with a graph of only bottom nodes, where the weight of a link between two nodes is determined by the amount of common surface nodes they are linked to in the bipartite graph.

Given the projection of the bipartite graph on the bottom nodes, we apply the hierarchical clustering method as described in Wichmann et al. (2020). Starting with the largest connected component in the bottom projection as the only cluster (which

represents the full global ocean), the clustering algorithm chooses one cluster at every iteration and splits it into two clusters, such that the Normalized Cut (NCut) is minimized (Shi and Jitendra, 2000). For $K$ clusters $S_1, \cdots, S_K$, the NCut is defined as:

$$\text{NCut}(S_1, \cdots, S_K) := \sum_{i=1}^{K} \frac{Q\left(S_i, S_i^C\right)}{Q\left(S_i, S\right)},$$

where $Q\left(S_i, S_j\right)$ is the sum of all weigths connecting $S_i$ and $S_j$, $S_i^C$ is the complement of $S_i$. By definition, the NCut increases

at every iteration (i.e. if the amount of clusters is higher).

We do not test the hierarchical clustering at 250 m day$^{-1}$ sinking speed of particles in the back-tracking analysis. The particle distributions spread less at this sinking velocity, and the bottom projection of the bipartite graph becomes disconnected.





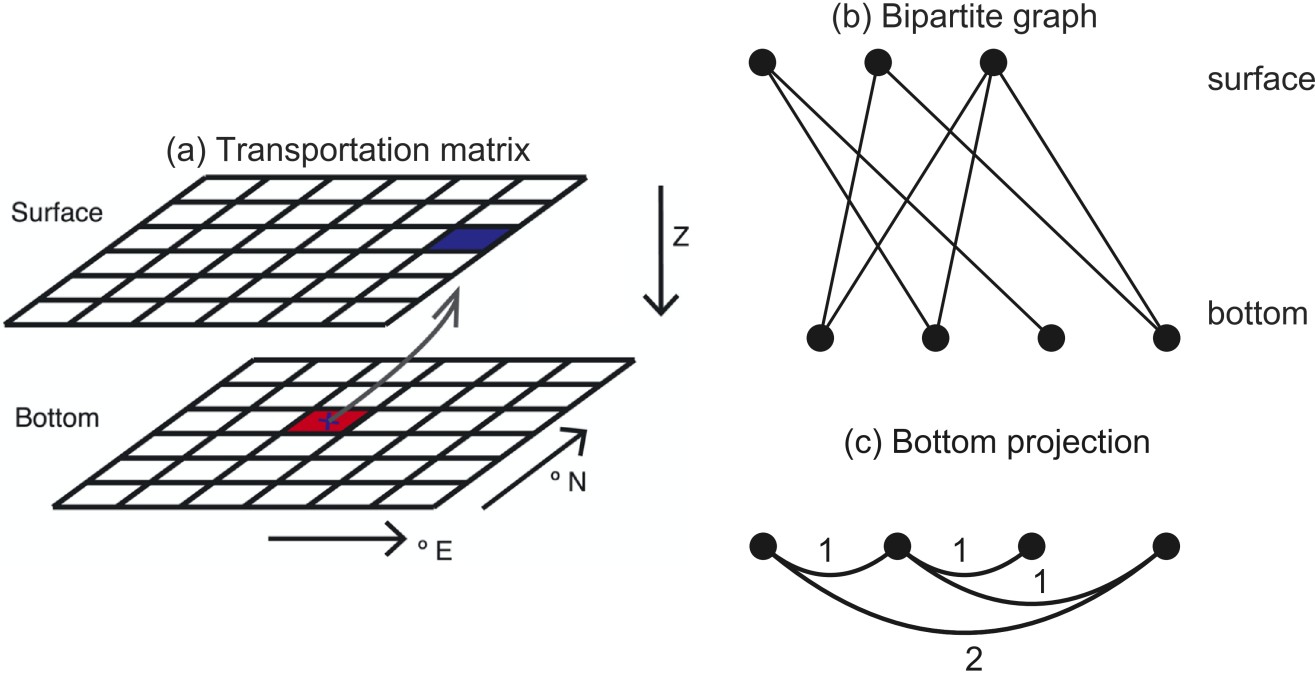

**Figure A1.** Illustrations for the embedding types that the hierarchical clustering method uses. (a) Illustration of the surface-bottom trans-portation matrix (figure adapted from Nooteboom et al. (2019)). The transportation matrix contains the probabilities that a particle that is found in a bottom box, originated from a surface box. The transportation matrix can also be interpreted as a bipartite graph in (b), which has a bottom projection (c): the bottom nodes are linked with a weight that is determined by the amount of mutually linked surface vertices in the bipartite graph.

Hence, these higher sinking speeds require a higher resolution binning of the input used, which exceeds given computational limitations.

## A2 OPTICS clustering

We use OPTICS to compare clusters with the sediment sample sites, which is density based and distinguishes different clusters from 'noise' (Wichmann et al., 2020). We apply OPTICS in this paper to the 'direct embedding' of surface origin distributions (Fig. 1a).

The main result from OPTICS is the reachability plot (see Wichmann et al. (2021) for more details). The reachability plot is a

representation of the global and local distribution of points (which represent sedimentary sites in this paper) at once. The valleys correspond to dense regions with similar surface origin location, while the hills correspond to the 'noisy' locations. The reacha-bility plot depends on a parameter $s_{min}$, for which we test multiple values ($s_{min} \in [100, 200, 300, 400, 500, 600, 700, 800, 900, 1000]$).
$s_{min}$ sets the minimum amount of 'nearby' points in the reachability plot for every point in a cluster (MinPts in Ester et al.

(1996)). In general, a larger $s_{min}$ results in a smoother reachability plot and larger clusters. If we let every particle release location (released on a $1° \times 1°$ grid) represent an area of 1 square degree ($\sim 10^4$ km$^2$), the OPTICS algorithm searches for a cluster with a spatial scale $\sim s_{min} \cdot 10^4$ km$^2$.

In this paper, we use $\xi$-clustering to obtain clusters from the reachability plot (Ankerst et al., 1999). This implies that we set a threshold on the steepness of the density ($\xi$), and cluster the valley of points that is surrounded by this steepness $\xi$. In general, a larger $\xi$ will reduce both the size and the amount of clusters.

**Appendix B: The distance matrices defined**

We use (symmetric) distance matrices based on four different metrics. First, we use a matrix that contains Euclidean taxonomic distances, calculated from the relative abundances (fractions) of species. Second, we use the absolute SST differences between the sites. Third, we use a distance which is based on the reachability from the OPTICS algorithm. Specifically, if $r(p_k)$ is the reachability of point $p_k$ and the $n$ points are ordered from $p_0 \cdots p_n$, the reachability distance between two sediment sample sites

(located near $p_i$ and $p_j$ respectively with $i \leq j$) is $D_{ji}^r = D_{ij}^r = \max_{i \leq k \leq j} r(p_k) - \min_{i \leq k \leq j} r(p_k)$. Intuitively, $D_{ij}^r$ represents how much one has to climb or descent in the reachability 'landscape,' if one likes to move from point $i$ to $j$. Fourth, we use a distance matrix which contains the spatial distance (in meters) between sediment sample sites. The partial Mantel test determines the correlation between the reachability distance and either SST or taxonomy distance matrices, keeping the spatial distance matrix constant.

*Author contributions.* PN designed and performed the research. AH, HD, ES and PB provided the research funding. PN prepared the manuscript with contributions from all authors.

*Competing interests.* The authors declare that they have no competing interests.

*Acknowledgements.* This work was funded by the Netherlands Organization for Scientific Research (NWO), Earth and Life Sciences, through project ALWOP.207. The use of SURFsara computing facilities was sponsored by NWO-EW (Netherlands Organisation for Scientific Re-
search, Exact Sciences) under the project 17189. PKB acknowledges funding from the European Research Council under the European Community's Seventh Framework Programme through ERC starting grant #802835 (OceaNice).



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
