# Peer review of "Sedimentary microplankton distributions are shaped by oceanographically connected areas"

_Earth System Dynamics, 2021_

## Author Comment (AC1)

**This is an important topic to better understand and potentially quantify the effect/amount of transported particles by ocean currents while sinking from the ocean surface to the sea-floor and forming an archive for paleorecontructions.**

**This process is hardly taken into account or discussed in the scientific reconstructing past ocean conditions which explains why the paper by [12] is hardly cited and surprisingly not taken up in the list of references in this manuscript. Please allow me to quote his first sentence from the abstracts:**

**'The interpretation of micropaleontological data based on the fossil remains of planktonic organisms requires an appropriate reference frame.'**

**Several papers have been published since then reporting observations and/or attempts to quantify this effect [2, 7, 8] where this particles are sometimes called expatriates. To my knowledge a paper modelling this effect and quantifying the consequence for the sediment composition is still missing, while much more articles concentrate of vertical mixing (bioturbation) in sediments, a process which changes the original composition of surface ocean sediments (including the expatriates).**

**An extreme effect of this kind of process on particles in the micro- to nano-scale is reported by [6] and the effect on establishing an age frame of marine sediments by radiocarbon dating.**

**This paper nicely clusters deep-sea sediment uses a 3D-flow model to shed more light on the complexity of the sedimentary microplankton composition.**

**Although not being an experts in statistics and modeling, the used methods sound carefully selected and applied.**

**The text is very well written and explanations are sound and convincing.**

We would like to thank Gerald Ganssen for his careful reading and his constructive comments.

Please find our replies and our proposed changes in the revised manuscript below.

On behalf of the authors,

Peter Nooteboom

**Changes in manuscript**

The papers [12, 6, 2] will be referred to in the introduction section and [12] will also be referred to in the discussion section of the new manuscript version.

**In the method section the authors do not mention the typical size fraction for their calculated sinking speeds of both dinoflagellate cysts and planktonic foraminifera; here it would be important to make a difference between empty shells and those still containing organic material which, during sinking, will get oxidized and the amount of gas within the shell will reduce sinking speed siginificantly.**
**A discussion on the potential effect of very slowly sinking particles reaching the sediment archive: How high do the authors estimate this bias?**

**Author's response**

In general, the processes that influence the sinking speed of marine particles are complex (in particular due to particle aggregation and fecal pelleting). Therefore, we do not test specific sinking speeds for different microplankton species. Instead, we use four constant sinking speeds (6, 11, 25, 250 m day$^{-1}$; see the Supporting Information), and find that the clustering structure is not sensitive to sinking speed, because of the similar spatially varying character of ocean advection that particles experience at these sinking speeds (L.174-178 and L.240-249 of the manuscript). Because the clustering structure is to first order independent of the sinking speed, it is irrelevant to add information on the size fractions and distinguish between empty shells and those that include organic material.

We consider 6 m day$^{-1}$ to be a low sinking speed for both planktic foraminifera and dinoflagellate cysts (dinocysts). We did not test lower sinking speeds, because the back-tracking analysis is computationally infeasible for 'very low' sinking speeds due to long particle travel times. Hence, we will not include a discussion on the effect of 'very low' sinking speeds on the clustering structure.

**Changes in manuscript**

We will add a sentences in the method section to explain why we do not test lower sinking speeds (L.59):
Sinking speeds lower than 6 m day$^{-1}$ can occur (e.g. due to oxydation of organic material and the development of gas within a shell), which may have an effect on the computed clusters. However, sinking speeds lower than 6 m day$^{-1}$ are not tested in this paper, because the backtracking method is computationally infeasible at lower sinking speeds due to long particle travel times.

**To further prove the applicability of the authors approach they should compare their results with selected case studies from literature, where lateral advection of sinking particles has been reported to contribute to the sediment association and might disturb the original surface ocean signal.**

**Author's response**

Comparison of the back-tracking analysis with specific case studies has already been done (e.g. [8, 11]). In order to rigorously compare our clustering results to other types of data than surface sediment sample sites, an extended dataset of microplankton at or near the ocean surface is required. For dinocysts, this dataset does not exist to our knowledge, because the

biological producers of dinocyst species (i.e. the dinoflagellate species) are often not known. For planktic foraminifera, the sediment trap data from [3] cannot be used, because most data is not from the near-surface, but from sediment traps at greater depths.

Hence, no dataset is available to make such a comparison.

**Changes in manuscript**

None.

**Finally:**
**I do not see how:**
**These type of studies could determine the relative contribution to the higher biodiversity outside compared to within oceanographically isolated clusters from ocean surface parameters, as well as dissolution [1, 10] and mixing of particles during their sinking journey. :**

**Author's response**

Microplankton biodiversity as measured in sediment sample sites is determined by (a) microplankton biodiversity near the ocean surface, (b) species-specific dissolution and (c) mixing of species during their sinking journey. Microplankton at sediment sample sites in oceanographically isolated clusters are likely less influenced by (c) compared to the 'noisy' areas. Hence, oceanographically isolated clusters can be used to determine areas where lateral transport does not influence the biodiversity in sedimentary sample sites.

**Changes in manuscript**

We rephrase this paragraph in the discussion (L.298):
'Fourth, our study provides micropalaeontologists with a tool to qualitatively assess the importance of lateral transport to sedimentary particle assemblages, which can be used in studies that compare measured biological diversity and environmental conditions in surface waters with their sedimentary remains (e.g. [3, 5]), particularly in those regions for which we here demonstrate noisy behaviour. These type of studies could determine the relative contribution to the higher biodiversity outside compared to within oceanographically isolated clusters from ocean surface parameters, as well as dissolution [1, 10] and mixing of particles during their : Within oceanographically isolated clusters, sedimentary microplankton biodiversity is only weakly determined by lateral particle transport compared to the microplankton biodiversity near the ocean surface and species-specific dissolution [1, 10] .'

**Detailed studies of productivity in surface mixing zones [9] probably might be much stronger than advection during the sinking of the particles. I hence would like the authors to more thoroughly explain this conclusion.**

**Author's response**

This is a good point, and in particular true for planktic foraminifera which are passively advected at the near-surface during their life span. A paragraph in the discussion section explains

this point (L.305-310).

Near-surface mixing of dinocysts is not an issue. Immediately after being produced, dinocysts start sinking passively. If dinoflagellates (the biological producer of the dinocyst) end up outside of their habitat due to surface mixing, they die, oxidize and do not end up in the sedimentary record.

**Changes in manuscript**

We will rephrase this paragraph to make this point clearer and add the reference [9] (L.305): 'The backtracking analysis on which we applied the clustering was designed for dinocysts, and not for foraminifera.  In particular, near-surface advection during the foraminifera life span may have a larger impact on its sedimentary distribution compared to the lateral transport during sinking [9]. Clustering results from this paper compared well with the foraminifera dataset in most cases, because the areas with strong particle mixing and lateral transport (i.e. their spatial dependence) are likely similar for foraminifera (and likely similar at the near-surface compared to other depth levels). Nevertheless, future work could apply these clustering methods on a backtracking analysis which is designed for foraminifera (similar to [11, 4]). This means that particles are released at the ocean bottom, tracked back in time until they reach the foraminifera dwelling depth, and finally tracked back during their life span at this dwelling depth.'

**References**

[1] Ivy Frenger, Matthias Münnich, and Nicolas Gruber. Imprint of Southern Ocean mesoscale eddies on chlorophyll. pages 4781–4798, 2018.

[2] Gerald Ganssen and Dick Kroon. Evidence for Red Sea surface circulation from oxygen isotopes of modern surface waters and planktonic foraminiferal tests. *Paleoceanography*, 6(1):73–82, 1991.

[3] Lukas Jonkers, Helmut Hillebrand, and Michal Kucera. Global change drives modern plankton communities away from the pre-industrial state. *Nature*, 372, 2019.

[4] Michael Lange and Erik van Sebille. Parcels v0.9: prototyping a Lagrangian Ocean Analysis framework for the petascale age. *Geosci. Model Dev. Discuss.*, (July):1–20, 2017.

[5] Julie Meilland, Hélène Howa, Vivien Hulot, Isaline Demangel, Joëlle Salaün, and Thierry Garlan. Population dynamics of modern planktonic foraminifera in the western Barents Sea. *Biogeosciences*, 17:1437–1450, 2020.

[6] G. Mollenhauer, T. I. Eglinton, N. Ohkouchi, R. R. Schneider, P. J. Müller, P. M. Grootes, and J. Rullkötter. Asynchronous alkenone and foraminifera records from the Benguela Upwelling System. *Geochim. Cosmochim. Acta*, 67(12):2157–2171, 2003.

[7] P D Nooteboom, P Delandmeter, Erik Van Sebille, Peter K Bijl, Henk A Dijkstra, and Anna S von der Heydt. Resolution dependency of sinking Lagrangian particles in ocean general circulation models. *PLoS One*, 15(9):1–16, 2020.

[8] Peter D Nooteboom, Peter K Bijl, Erik van Sebille, Anna S von der Heydt, and Henk A Dijkstra. Transport Bias by Ocean Currents in Sedimentary Microplankton Assemblages : Implications for Paleoceanographic Reconstructions. *Paleoceanogr. Paleoclimatology*, 34, 2019.

[9] Janneke J Ottens and Alexandra J Nederbragt. Planktic foraminiferal diversity as indicator of ocean environments. *Marine Micropaleontology*, 19:13–28, 1992.

[10] Ben J Taylor, James W B Rae, William R Gray, Kate F Darling, Andrea Burke, Rainer Gersonde, Andrea Abelmann, Edith Maier, Oliver Esper, and Patrizia Ziveri. Distribution and ecology of planktic foraminifera in the North Pacific: Implications for paleo-reconstructions. *Quat. Sci. Rev.*, 191:256–274, 2018.

[11] Erik van Sebille, Paolo Scussolini, Jonathan V Durgadoo, Frank J C Peeters, Arne Biastoch, Wilbert Weijer, Chris Turney, Claire B Paris, and Rainer Zahn. Ocean currents generate large footprints in marine palaeoclimate proxies. *Nat. Commun.*, 6:6521, 2015.

[12] P K Weyl. Micropaleontology and ocean surface climate. *Science*, 202:475–481, 1978.

---

## Author Comment (AC2)

**Nooteboom et al., use a strongly eddying global ocean model simulation to investigate the influence of particle advection by ocean currents on sedimentary microplankton distributions.**
**They show that the effect of clustering by the advection that leads to regions that originate from clusters and regions that are more noisy (originate from various source positions) can be detected in the microplankton distributions of dinocyst and foraminifera assemblages.**

**This is a very interesting study, contributing a better understanding of sedimentary microplankton distributions that are routinely used to reconstruct past climate conditions. However, in its present form, the manuscript is quite challenging to read (for a person from a (paleo)climate background) and while I agree with the conclusions that an effect of the advection can be detected in the assemblages; even after reading it several times, it is unclear to me how strong this effect is.**
**As the metrics used are rather complex and unfamiliar to me (e.g. Partial Mantel correlation of the reachability distance D, reachability estimated with the OPTICS algorithm, with the taxonomy with spatial distance held constant) it is challenging to judge the robustness of the results.**
**Overall, I recommend publication after clarifications are made according to the comments below.**

We would like to thank the reviewer for the careful reading and the constructive comments.

Please find our replies and our proposed changes in the revised manuscript below.

On behalf of the authors,

Peter Nooteboom

**Oceanographically disconnected clusters and ANOSIM results**

**Figure 2 shows that the sedimentary microplankton composition are more similar within than between clusters. However, as the authors note themselves, this could also be explained with the fact that sediment sites within clusters are closer to each other. Thus, to the reader, it is unclear whether this result indicates any effect of the advection connections on the assemblages. Would it be possible to create a proper null hypothesis for this experiment e.g. by creating random clusters with the same size as the true clusters and test if the similarity in the true clusters is higher than in the surrogate clusters?**

**Author's response**

The ANOSIM [6, 7] test shows that sediment sites within clusters are more similar compared to sediment sites between clusters. ANOSIM already tests the significance of these results and shows that p-value<0.01 if clusters are not very large (i.e. after a few iterations of the clustering method). This implies that the results are significant compared to using random clusters. We are not aware of a method that rigorously controls for the distance effect in these clusters. Extensive null-hypothesis tests on synthetic data is provided by

**Changes in manuscript**

None.

**Oceanographically isolated clusters**

**According to the authors, Figure 3c demonstrates the effect of the reachability / isolation on the assemblage. However, despite the explanations, I find it challenging to understand and interpret this diagnostic. If I understand it right, it compares reachability distance (which is only defined in the Appendix) to the distance in SST and distance in taxonomy while removing the effect of the geometrical/spatial distance. Maybe the authors can explain in a simple way what this means.**

**Author's response**

This is a correct interpretation of the result. The definition of the reachability is provided by [1].

**Changes in manuscript**

We will change the caption of figure 3, such that the meaning of the reachability distance is clearer:
'Partial Mantel correlation of the reachability distance $D_r$ (a lower value of $D_r$ between two sites indicates a stronger oceanographic connection between these sites; see appendix B) with the taxonomy (red) and SST (black), both with spatial distance held constant, for different $s_{min}$ values.'
We will add a sentence in the main text that better explains this result:
'The reachability distance ($D_r$; see Appendix B) between sediment sample sites correlates

positively with sediment sample taxonomy. Furthermore,  this correlation is independent of the spatial distance between sites, according to the partial Mantel tests [8](Fig.3c). This means that oceanographically connected sites have a similar taxonomy, independent of their spatial distance.'

**Why is the distance in reachability important for the distance in assemblages and not the reachability itself. I would have expected a strong reachability (thus a connection to many sites and thus many environmental conditions) to lead to a diverse taxonomy but it is less clear why the distances in these parameters should be related.**

**Author's response**

We used the reachability distance here, because a partial Mantel test requires distance measures between sites in order to calculate correlations between variables while controlling for other variables (as is explained in the method section; L102-103). The reachability itself is not a distance measure.

Furthermore, it is not only the reachability itself, but also the ordering of points/sites as shown in Figs. 3c and 4c that determines whether two sites are oceanographically connected or not. For instance, as one can see in Fig. 4c, some sites could have a similar reachability value, while being located in a different oceanographically isolated cluster (hence these sites are not oceanographically connected). Therefore, the reachability values themselves do not indicate whether two sites are oceanograpically connected to each other.

The observation that sites with a high reachability are likely to have a more diverse taxonomy is correct and we investigate this effect later in the manuscript (figure 7).

**Changes in manuscript**

None.

**If one looks at the distances for some reason, intuitively I would have thought that the distance in taxonomy would have to be compared to the distance in SST and then compared to the distance in reachability?**

**Author's response**

This paper is about the oceanographic connection between sites, and how this shapes the microplankton composition in these sites. Therefore, it is in particular the correlation between reachability distance and taxonomic distance that is of importance here, and intuitive to present in this figure.

We also present the correlation between SST and the reachability distance to show that the correlation between SST and the reachability distance is only weak. Hence the correlation between the reachability distance and the taxonomic distance is not caused indirectly by both a strong correlation between SST and taxonomy (which is known to be the case) and a strong correlation between SST and $D_r$. As is explained in the main text (L.196-198), this may be the case for low values of $s_{min}$ ($s_{min} \leq 200$).

**Changes in manuscript**

None.

**In Figure 3c, there is a strong correlation of reachability distance to SST for forams for small smin; is this discussed in the text? I might have missed it.**

**Author's response**

This result is discussed in the manuscript (see L.196-198).

**Changes in manuscript**

None.

**Finally, when I try to visualize what is going on, I look at the reachability in Figure 3a but unfortunately, just at the parameter smin=300 used in Figure 3a the correlation in Figure 3c is nearly zero.**

**Author's response**

The main reason that we use $s_{min} = 300$ here, is that we also keep using this $s_{min}$ value in later figures. It is true that the correlations are relatively low at $s_{min} = 300$, but the correlation is statistically significant at this scale, and likely not (or only little) influenced by the low correlation between taxonomic and SST distance which is close to zero.

**Changes in manuscript**

We add a figure in the supporting information which is the same as Fig. 3, but with $s_{min} = 500$ in subplot (a), (b) (Fig. 1 at the end of this document).

**Line 192 writes that the reachability distance is independent of the spatial distance between sites, but isnt the effect of the spatial distance removed/ controlled for in the partial Mantel correlation that is used here? Likely these are all quite ignorant and stupid comments, but they might show the challenge in understanding this part of the paper.**

**Author's response**

If the Mantel correlation is significantly positive, while the effect of spatial distance between the sites is removed, then this correlation is independent of the spatial distance. Hence, it is the correlation between the reachability and taxonomy (not the reachability itself) which is independent from the spatial distance between sites.

**Changes in manuscript**

We will add this explanation to the method section (L.103):
'A partial Mantel test requires at least three types of distance matrices, which contain distances between the sediment sample sites. We calculate the Mantel correlation between taxonomic

distance and a distance which is determined from the reachability of the OPTICS clustering (see Appendix B), while we  remove the effect of the spatial distance between sites on this correlation. If the Mantel correlation is significantly positive between these variables while removing the effect of the spatial distance metric between the sites, then this correlation is independent of the spatial distance. '

L192 will also be changed:
'The reachability distance ($D_r$; see Appendix B) between sediment sample sites correlates positively with sediment sample taxonomy. Furthermore,  this correlation is independent of the spatial distance between sites, according to the partial Mantel tests (Fig.3c). This means that oceanographically connected sites have a similar taxonomy, independent of their spatial distance.'

**In Figure 4, a dimension reduced version of the species composition is compared to the OPTICS clusters and it is argued that spatially closed clusters (e.g. red and yellow) show a well separated taxonomy, arguing for an influence of the current-shaped clusters. But arent the surface conditions as SST also very different for the regions of the red and yellow clusters the clusters are separated by the Antarctic Polar Front and thus even in a classical interpretation of the taxonomy only driven by e.g. temperature, we would expect a separated taxonomy?**

**Author's response**

Indeed, these clusters are separated by the Antarctic Polar Front (APF), which roughly coincides with the Antarctic Circumpolar Current (ACC) that is responsible for the oceanographic isolation of these clusters. Moreover, SST shapes part of the microplankton composition in these clusters.
However, when clustering all sediment sites, it is difficult to find clusters that only include sites with a clearly distinct microplankton taxonomy compared to other clusters. A good example is the sediment site in Fig. 1b that is indicated with the diamond (i.e. the 'noisy' site). This sediment site is closely located to cluster 1 in the same figure, but contains more species compared to the site in cluster 1. If 'noisy' sites like these are included in clusters, it will likely result in overlapping/non-distinct clusters in Fig. 4. This means that the oceanographic connection, and not SST, between the sites within clusters makes their taxonomic composition distinct from sites outside of the clusters.

**Changes in manuscript**

To make this point clear, we will add a sentence in the main text (L.120):
'The comparison between the clusters and taxonomic distance of Southern Hemisphere sample sites in these clusters (Fig. 4c and 4d) becomes interesting for clusters which are spatially close (Fig. 4b). For instance, the red and yellow cluster in the South-Atlantic Ocean are spatially close, but sediment samples in those clusters are separated by their observed dinocysts taxonomy (Fig. 4c). This implies that we find a signal of the oceanographic separation of these areas in the sedimentary data. If noisy sites (such as the noisy site in Fig. 1b) would be part of clusters, sites in different clusters are likely to contain a similar microplankton composition and the taxonomic separation of clusters is unclear. '

**Figure 5 shows the relation between microplankton species variability and environmental variables in clustered and noisy sites. This demonstration is easy to follow and shows an increase in the explained variance of 5-15% when excluding the noisy sites.**

**Again, Im not yet fully convinced that this must be related to the 3D advection / clustering. If we expect that there is a correlation of the spatial distance and the taxonomy (e.g., due to some secondary spatially varying variable); wouldnt we expect that picking some spatial patches /subsets from the full field would remove noise and increase the explained variance? One possibility to disprove this hypothesis would be to repeat the same analysis with similar sized randomly spaced clusters.**

**Author's response**

As indicated at the end of the figure 5 caption, we applied such a hypothesis test (specifically a so-called permutation or randomnization test; see the method section for a detailed explanation) and found low p-values ($<0.0001$ and $0.024$ for dinocysts and foraminifera, respectively), which indicates that these results are statistically significant.

**Changes in manuscript**

None.

**(Line 231 ff) Finally, it is argued that the clustered samples are less taxonomically mixed; Maybe I missed it, but can it be excluded that the surface conditions (as SST, Nitrate) are not just less variable inside the clusters than outside (1. Because of the spatial distance inside and outside of the clusters; 2. As the clusters avoid the fronts?).**

**Author's response**

Indeed, it could be that environmental conditions that are favor a high surface biodiversity correlate with oceanographic connectivity. In this case, the relation between oceanographic connectivity and sedimentary biodiversity may be indirect . This point is described in the discussion section (L.300-304).

**Changes in manuscript**

None.

**Line 265ff: Implications**

**Author's response**

This will be corrected.

**Changes in manuscript**

We will change the sentence (L.265): 'These provinces have  implications for sedimentary microplankton assemblages.'

**These are important results for the paleoclimate community and maybe some clarifications could help to increase the impact of the study for this field. Do I understand it right that the recommendation would be to only train the transfer function inside a cluster (or pick the analogues) inside a cluster?**

**Author's response**

Indeed, these clusters can be used to only train a transfer function inside a cluster.

**Changes in manuscript**

We will change a sentence in the discussion section, such that this suggestion becomes clearer (L.277):
'The hierarchical clustering method (which finds oceanographically disconnected clusters) can help to determine bounds on the spatial extent that is used for the training of transfer functions (e.g. a transfer function can be trained on sites within a single cluster), since it shows areas which are oceanographically separated from each other.'

**Line 285: How could the provinces be used to correct for ocean connectivity in assimilation approaches? By using the modern clusters?**

**Author's response**

A challenge in this type of data assimiliation, is that the spatial distribution of sites is heterogeneous [3]. However, sites can be located close together and be oceanographically disconnected at the same time, or they can be located far away from each other and still be oceanographically connected. Hence, the spatial distribution of areas that proxies represent may be different from the distribution of sites themselves. The clusters in this paper may be of help to set up a 'drifting reference frame' [4]. It is out of scope for this paper to implement a correction for oceanographic connectivity in these data assimilation approaches.

For data assimilation approaches in past climates, it may be necessary to determine these clusters in the studied time period, possibly with the use of palaeoceanographic model simulations (at sufficiently high spatial resolution).

**Changes in manuscript**

We will change the sentence in the discussion section (L.284):
' These provinces can be used to correct for ocean connectivity by providing a different reference frame [4] if the proxies are used to assimilate e.g. global sea surface temperature fields (as in [3]). This may require the computation of these clusters in the past using palaeoceanographic models.'

**Line 287: Why should the disconnected provinces provide the spatial structure for proxy calibration parameters? Variations in the proxy calibration parameters could be either due to secondary variables not considered in the calibration, or due to different species**

**/ variants of the organisms recording the climate signal. Why would either of these options follow the disconnected provinces instead of following the climatic / oceanographic conditions?**

**Author's response**

As described in [2], proxy calibrations may have strong regional differences in their residuals. In [2], these high residuals are attributed to either lateral advection or specific regional processes that influence the proxy-environment relationship (i.e. the effects of 'secondary variables' on proxy calibrations are spatially varying).

Sedimentary particles in clusters share similar near-surface origin locations. First, this means that if sites in a cluster are greatly influenced by lateral advection, then all of these sites are biased in the same direction and with a similar magnitude. Second, it means that the sites are likely to share similar processes that determine the proxy-environment relationship. The clusters in this manuscript could therefore provide the spatial structure that is used for proxy calibration.

It is out of scope for this paper to provide such a calibration which uses the provinces to correct for high regional residuals in proxy calibrations.

**Changes in manuscript**

We will change the sentence in the discussion section, such that it explains clearer why the clusters in this paper can be used in proxy calibrations (L.287):

' Since proxy calibration residuals are often high in specific areas and related to lateral advection [2], the (dis)connected provinces from this paper can provide a spatial structure that such a regression model uses for core-top calibration.'

**In summary, I recommend strengthen and clarifying the argumentation for a strong effect of the 3D advection / clustering on the assemblages. Depending on the outcome, I suggest formulating the statements of the advection effect a bit more moderate. This applies especially to the title Sedimentary microplankton distributions are shaped by oceanographically connected areas . A weaker version would be e.g.. microplankton distributions are influenced And line 8 These provinces explain the microplankton composition, together with e.g. ocean surface environmente.g. to in addition to the ocean surface environment, the provinces contribute to the microplankton composition.**

**Author's response**

We hope that our replies and suggested changes in the manuscript will convince the reviewer that 3D advection plays an important role in shaping sedimentary microplankton composition.

**Changes in manuscript**

We will change line 8:

'We find that these provinces can be detected in global datasets of sedimentary microplankton assemblages, demonstrating the effect provincialism has on the composition of sedimentary remains of surface plankton. These provinces explain the microplankton composition,

 in addition to e.g. ocean surface environment.'

**Minor comments:**

- **In some parts, only sites in the SH are considered (L.50 in order to limit the total diversity of microplankton species). Why is a high diversity an issue?**

**Author's response**

A high species diversity within the full dataset is in particular an issue when visualising species composition in two dimensions using dimensionality reduction methods such as multidimensional scaling and canonical correspondence analysis (used in Figs. 4, 5 and 6). Therefore, we only used Southern Hemisphere (SH) sites in these figures.

**Changes in manuscript**

None.

- **The figure captions are difficult to read, and I see room for improvement here. I would suggest starting the caption with an overview sentence; than describe the panels and then potentially draw some conclusions.**

**Author's response**

We decide to keep the figure captions descriptive and will not include conclusions in the figure captions, as the ESD submission guidelines suggest.

**Changes in manuscript**

We do add a panel description to the figure 7 caption:
'Sedimentary microplankton biodiversity outside minus inside  oceanographically isolated provinces. The average Shannon entropy of (a) dinocyst and (b) foraminifera sediment samples inside OPTICS clusters $\overline{N}_s^c$ compared to outside clusters $\overline{N}_s^{nc}$ for different values of $s_{min}$ (i.e. the minimum size of clusters) and $\xi$ (i.e. the level of isolation). High values indicate that the number of species in samples within clusters are lower and species are distributed less evenly in samples compared to samples outside clusters. Blue are configurations of $s_{min}$ and $\xi$ for which no sediment sample sites are part of a cluster.'

- **Figure 3a. (a) Scatter plot of the site reachability in space; I wouldnt call this a scatter plot.**

**Author's response**

We will this in the new manuscript version.

**Changes in manuscript**

We will remove 'scatter plot' from the caption: ' Site reachability in space: sites in dense areas with a low reachability are oceanographically isolated.'

- **Figure 3 and 4: Sediment location this is a scalar number of a 2D location; I guess nearby records have a similar location index but its unclear how this is defined**

**Author's response**

Sediment locations are scalar numbers of 2D locations. However, in Figs. 3b and 4b these sites are ordered in only one dimensions by the Ordering Points To Identify Clustering Structure (OPTICS) algorithm. Hence, nearby records do not necesarily have a similar location index. For the details about the OPTICS algorithm we refer to the appendix and several references (e.g. [1, 5]).

**Changes in manuscript**

None.

**References**

[1] Mihael Ankerst, Markus M Breunig, Hans-Peter Kriegel, and Jörg Sander. OPTICS : Ordering Points to Identify the Clustering Structure OPTICS. *SCM Sigmod Rec.*, 1999.

[2] Jessica E Tierney and Martin P Tingley. BAYSPLINE: A New Calibration for the Alkenone Paleothermometer. *Paleoceanogr. Paleoclimatology*, 33:281–301, 2018.

[3] Jessica E Tierney, Jiang Zhu, Jonathan King, Steven B Malevich, Gregory J Hakim, and Christopher J Poulsen. Glacial cooling and climate sensitivity revisited. *Nature*, 584, 2020.

[4] P K Weyl. Micropaleontology and ocean surface climate *Science*, 202:475-481, 1978.

[5] David Wichmann, Christian Kehl, Henk A Dijkstra, and Erik Van Sebille. Ordering of trajectories reveals hierarchical finite-time coherent sets in Lagrangian particle data : detecting Agulhas rings in the South Atlantic Ocean. *Nonlinear Process. Geophys.*, 28:43–59, 2021.

[6] Robert K. Clarke. Comparisons of dominance curves. *Journal of Experimental Marine Biology and Ecology*, 138:143-157, 1990.

[7] Robert K. Clarke. Non-parametric multivariate analyses of changes in community structure. *Aust. J. Ecol.*, 18:117-143, 1993.

[8] Legendre, P. and Legendre, L.: Numerical Ecology, Elsevier, Amsterdam, 3rd edn., 2012.

[Figure]

Figure 1: Same as figure 3, but with $s_{min} = 500$.